# Iron Deficiency in Tomatoes Reversed by *Pseudomonas* Strains: A Synergistic Role of Siderophores and Plant Gene Activation

**DOI:** 10.3390/plants13243585

**Published:** 2024-12-22

**Authors:** Belén Montero-Palmero, Jose A. Lucas, Blanca Montalbán, Ana García-Villaraco, Javier Gutierrez-Mañero, Beatriz Ramos-Solano

**Affiliations:** Plant Physiology, Pharmaceutical and Health Sciences Department, Faculty of Pharmacy, San Pablo—CEU Universities, 28668 Boadilla del Monte, Spain; mariabelen.monteropalmero@ceu.es (B.M.-P.);

**Keywords:** chlorosis reversion, iron nutrition, photosynthesis, Plant Growth Promoting Bacteria (PGPB), *Pseudomonas*, siderophores

## Abstract

An alkaline pH in soils reduces Fe availability, limiting Fe uptake, compromising plant growth, and showing chlorosis due to a decrease in chlorophyll content. To achieve proper Fe homeostasis, dicotyledonous plants activate a battery of strategies involving not only Fe absorption mechanisms, but also releasing phyto-siderophores and recruiting siderophore-producing bacterial strains. A screening for siderophore-producing bacterial isolates from the rhizosphere of *Pinus pinea* was carried out, resulting in two *Pseudomonas* strains, Z8.8 and Z10.4, with an outstanding in vitro potential to solubilize Fe, Mn, and Co. The delivery of each strain to 4-week-old iron-starved tomatoes reverted chlorosis, consistent with enhanced Fe contents up to 40%. Photosynthesis performance was improved, revealing different strategies. While Z8.8 increased energy absorption together with enhanced chlorophyll “a” content, followed by enhanced energy dissipation, Z10.4 lowered pigment contents, indicating a better use of absorbed energy, leading to a better survival rate. The systemic reprogramming induced by both strains reveals a lower expression of Fe uptake-related genes, suggesting that both strains have activated plant metabolism to accelerate Fe absorption faster than controls, consistent with increased Fe content in leaves (47% by Z8.8 and 42% by Z10.4), with the difference probably due to the ability of Z8.8 to produce auxins affecting root structure. In view of these results, both strains are effective candidates to develop biofertilizers.

## 1. Introduction

In plant nutrition, the limited availability of one essential element will determine harvest yield. Iron (Fe) is an essential element that becomes unavailable for plants in calcareous and alkaline soils, compromising plant growth. The main visual Fe deficiency symptoms in plants are leaf yellowing or chlorosis, due to a decrease in chlorophyll content, as it is an essential cofactor for its synthesis, and iron does not translocate to other organs; hence, iron deficiency shows preferably in new leaves [1]. Moreover, when chlorophyll content diminishes, light energy harvesting will be compromised, and this is primarily reflected in cellular sugar metabolism, in molecular reprogramming for stress adaptation, and at the end, in a low crop yield [2].

Iron absorption mechanisms by dicotyledonous roots are well established [1]. Upon sensing Fe deficiency, plants attempt to solubilize Fe^3+^ by acidifying the rhizosphere, releasing organic acids and/or protons (H^+^). This latter strategy involves activating plasma membranes’ H^+^-ATPase pumps (HAs) to lower the pH and solubilize Fe^3+^ in rhizosphere soil. Subsequently, the Fe-reductase (FRO) reduces soluble Fe^3+^ to Fe^2+^, and the Fe^2+^-carrier (IRT) will transport reduced iron through the plasma membrane into the cytosol [3]. Releasing carboxylic acids and phenolic compounds to the rhizosphere is another strategy, as these molecules behave as phyto-siderophores to improve the solubility of poorly accessible Fe [4]. Moreover, under Fe scarcity in soils, plants recruit siderophore-producing beneficial bacterial to improve Fe^3+^ chelation, making it possible for the Fe-reductase to proceed [5]. In addition to improving plant nutrition, the beneficial effects of Plant Growth Promoting Bacteria (PGPB) not only depend on the direct effect of bacterial siderophores chelating Fe, but also on the potential ability of PGPB to induce plants’ gene expression, thus enhancing the plant fitness [6]. This significant effect on plant growth promotion may involve indirect mechanisms like iron chelation, nutrient mobilization [7], antifungal chitinases [8], or direct mechanisms, involving plant gene reprogramming and resulting in enhanced adaptation to biotic or abiotic stress, or alterations of plant hormonal balances, which may directly affect plant growth and development [9,10,11].

As low Fe availability in alkaline soils limits crop yield, agricultural fertilizers with sulphate, EDDH-like, or organic Fe-chelated compounds have been traditionally used [12]. This classical fertilization ensures absorbable Fe delivery to plants, but does not improve the solubilization of Fe naturally present in soils. As organic farming calls for more natural and efficient practices, traditional Fe fertilization is still to be replaced. In line with this attempt, fertilization with PGPB appears as a promising alternative, for their biostimulating and biofertilizing properties, improving nutrient assimilation and crop quality [13]. Consequently, fertilizers formulated with siderophore-producing PGPB in soils poor in soluble Fe are good candidates to increase Fe nutrition without further additions. Hence, the challenge lies in the isolation of effective candidates to improve iron nutrition. To achieve this purpose, the best strategy is to use the plants’ ability to select bacterial strains that best fit their needs for survival [14]. In line with this statement, *Pinus pinea*, a mediterranean plant that usually grows in sandy nutrient-poor soils, has proven to be excellent at selecting isolates [15], which resulted in several strains being effective for plant growth [16,17] and enhanced defence against pathogens [18].

In this context, the aim of this study was to identify bacterial strains able to improve plant nutrition, focusing on iron nutrition in alkaline soils for the further development of biofertilizers. For that purpose, two actions were taken: (i) selecting siderophore-producing bacterial isolates from a large subset of isolated PGPB by sequencing 16S rDNA, and (ii) evaluating the ability of the best two siderophore-producing *Pseudomonas* sp. strains (Z8.8 and Z10.4) to revert chlorosis in Fe-starved tomato plants. For that purpose, tomato plants were Fe-starved and bacterial treatments were delivered several times, with an oxidized Fe source. Photosynthesis and chlorosis symptoms were evaluated, as well as pigments, nutrient contents and the expression of genes involved in Fe absorption.

## 2. Results

The 210 isolates from a bacterial screening assay carried out in the *P. pinea* rhizosphere were characterized at the morphological level (i.e., morphology, Gram-staining, and sporulation ability). The totality of the isolates were rod-shaped, most of them were Gram negative (79%), and among the Gram-positives, 56% of them were sporulating bacilli (Appendix A). Then, all isolates were characterized by their putative plant growth-promoting traits in vitro (siderophore production, phosphate solubilization, auxin production, and chitinases producers). A notable proportion of those strains was able to solubilize iron (70%) and phosphate (46%), being much lower the number of bacteria able to produce chitinases (2%). Among all isolates, 17% were able to produce auxin-like compounds, this trait being less represented in the whole community as it represents only 10% of the siderophore producers.

The 60 strains producing halos larger than 3 mm after 48 h in CAS plates were selected for further study, checking their ability to solubilize Mn^2+^ or Co^+^ (Appendix A). All 60 isolates were also good solubilizers of Mn^2+^ and Co^+^, producing halos over 5 mm. Despite none of the selected isolates being able to produce chitinases, phosphate-solubilizers (19) and auxin-producing isolates (18) were represented (Table 1). Among the 60 isolates, four strains were outstanding (Z7.38, Z7.40, Z9.3, and Z9.32) as they showed all the in vitro abilities tested (Appendix A).

After performing phylogenetic analyses with the 16S rRNA sequences, there were three different groups. One of them had only one strain identified as *Chryseobacterium*, the second group gathered two *Bacillus* strains, and the third and most abundant group included fifty-seven strains belonging to the *Pseudomonas* genus (Figure 1). The GenBank accession numbers of the 16S rRNA gene sequences from the 60 isolated strains are listed in the Appendix A.

Two strains were selected from the most abundant group: *Pseudomonas* Z8.8 and Z10.4, since they showed the best ion-mobilization potential. *Pseudomonas* Z10.4 was the greater siderophore producer, making the largest halos (9 mm in Fe^3+^, 11 mm in Mn^2+^, and 8 mm in Co^2+^; Table 2) and *Pseudomonas* Z8.8 was also able to produce auxins in vitro. Therefore, these strains were selected for biological assay to revert iron chlorosis.

To validate the onset of iron deficiency, the physiological parameters of control plants were compared with the positive control. Photosynthesis was affected by experimental conditions, with a significant decrease in the effective quantum yield efficiency (FPSII) and a significant increase in non-photochemical quenching (NPQ) with regards to positive controls (Figure 2A). Maximum potential photosynthetic efficiency (Fv/Fm) was not affected. In addition to yellowing, a significant decrease in photosynthetic pigments content was observed (Figure 2B).

Analysis of leaves showed a lower concentration of all inorganic nutrients determined, as compared to positive controls (Table 3), these differences being significant for Fe, Mn, and B.

Once nutrient starvation conditions were confirmed (Table 3), inoculated plants were analyzed. A remarkable positive effect of the bacterial treatments on plant survival was detected. In controls, only 43% of the non-inoculated plants survived, while it rose up to 79% in Z8.8-inoculated plants and 100% in Z10.4-inoculated plants, which were also more vigorous. Moreover, the number of non-chlorotic plants increased up to 43% in Z8.8-treated plants, while the control barely reached 18% (Figure 3).

Nutrient contents were improved by bacterial inoculation. Iron content improved by 47% (Z8.8) and 42% (Z10.4) compared to control (Table 4). Not only the Fe acquisition improved, but Mn assimilation was also significantly increased due to inoculation with *Pseudomonas* Z8.8. No significant increments in B, Co, Zn, Al, or Cu contents were registered, but a significant decrease in Cl in plants treated with both *Pseudomonas* strains was recorded (Table 4).

In Fe-starved plants, Fv/Fm was under the physiological threshold (0.8 relative fluorescence units, RFU, Figure 4A), while PGPB-treated plants showed a significant increment up to optimal levels. A significant increase was also detected in FPSII with both strains (Figure 4B). However, the NPQ was affected differently by each strain; while plants inoculated with Z8.8 showed a remarkable and significant increase in NPQ, Z10.4-inoculated plants did not modify NPQ with respect to control (Figure 4C).

Photosynthetic pigments also showed a specific response to each strain (Figure 5). Chlorophyll “a” was significantly accumulated in Z8.8-treated plants (up to 25%) compared with the non-treated controls, while it was significantly reduced in plants treated with Z10.4 (40% less). Moreover, chlorophyll “b” and carotenoid concentrations were significantly reduced only in Z10.4-inoculated plants, and were not affected by *Pseudomonas* Z8.8.

Finally, the expression of Fe deficiency-responsive genes (*plasma membrane H^+^-ATPase 1*, *HA1*; *Fe (III) chelate reductase*, *FRO*; and *iron Fe (II) root transporter*, *IRT*) was analyzed, and expressed as the number of transcripts (differential expression, M) accumulated in inoculated roots in comparison with the expression levels in non-inoculated roots. Transcript accumulations of *HA1* and *FRO* were significantly lower in Z8.8-inoculated plants, and expression of *IRT* was significantly lower in Z10.4-inoculated plants (Figure 6). Moreover, the data show a tendency of the progressive repression of *HA1*, at first place, then *FRO*, and finally *IRT* (Figure 6).

## 3. Discussion

Based on this study’s results, using siderophore-producing PGPB is a feasible tool to alleviate iron starvation in plants growing in alkaline soils. Consequently, selecting effective PGPB with this trait is of upmost importance for agriculture. To achieve this goal successfully, scientists have taken advantage of the plant’s ability to select strains that best contribute to plant fitness and survival, directing screening efforts at plants that usually live under harsh conditions [14,19]. In this sense, PGPB associated with coniferous tree species living in extreme conditions, with a long life cycle, have provided effective PGPB strains for inoculants [10]. The results presented here support this capacity, as iron and phosphate solubilization appear as the most frequent traits. Moreover, auxin-like producers contribute to enhanced nutrient absorption by increasing the root surface, resulting in a coordinated mechanism to further improve plant growth [20]. Consistent with the plant’s demands to cover urgent needs, nutrient-related traits are far more frequent than chitinase producers, suggesting a low incidence of fungal diseases in the area (Table 2).

Moreover, PGPB need to be effective in different plant species in order to qualify for marketable biofertilizers. Consistently, the two in vitro siderophore-producing *Pseudomonas* (Z8.8 and Z10.4) strains isolated from the nutrient-poor pine rhizosphere, followed by an in vivo validation in tomato plants, confirm their broad spectrum under greenhouse conditions.

Increasing nutrient solubility is a critical step for plant nutrition [21]. To overcome this situation, plants release ferric iron-specific ligands and phyto-siderophores to increase the solubility of ferrous ions and facilitate the further steps in the root epidermal entrance of Fe II and other nutrients [6], especially divalent cations such as Zn [22]. In addition to plant phyto-siderophores, siderophore-producing PGPB participate in the net siderophore balance in the rhizosphere, contributing to increased solubilization of Fe for the plant, as supported by the increased Fe content on Z8.8 and Z10.4 treated plants (Figure 3). As a consequence, these strains can be included among the *Pseudomonas* candidates with the ability to enhance agricultural food crop fitness [23]. The better performance of *Pseudomonas* Z8.8 on reverting chlorosis may be due to its ability to produce auxin-like compounds that may affect plants’ hormone balance, and thus contribute to a potential increase in root development, improving water and nutrient absorption, as well as solubilization of other nutrients such as phosphate [13].

However, these analyses also show that there is not a direct relationship between the in vitro and in vivo bacterial abilities; the amounts of Mn and Co in tomato leaves are not consistent with the in vitro ability of Z8.8 and Z10.4 strains to mobilize these elements (Table 4). The difference may be due to the low availability of these nutrients on the substrate, or interactions in the absorption processes in the assayed conditions [21]. Therefore, in vivo analyses should always be made to confirm the PGPB capacity.

Around 45% more Fe was accumulated in the leaves of inoculated plants, consistent with similar studies on other bacterial genera like *Chryseobacterium* [24], *Azospirillum*, *Gluconacetobacter* [25], or *Bacillus* [26]. In addition to the improved Fe assimilation registered in iron-starved tomato plants treated with both siderophore-producing strains, plants showed a better physiological condition, highlighting the multiple benefits contributing to plant fitness induced by PGPB [9]. On the one hand, the recovery of photosynthetic efficiency (FPSII) to optimal levels in plants treated with both *Pseudomonas* indicates a significant improvement in light energy assimilation (Figure 5). Strain Z8.8 increased the chlorophyll “a” concentration, consistent with the increase in energy absorption potential (Fv/Fm) [27] and energy dissipation (NPQ) to prevent a massive ROS production [28,29]. Conversely, Z10.4 decreased photosynthetic pigment concentration (Figure 6), suggesting a more conservative strategy in energy assimilation, also contributing to photoprotection [9]. The conservative strategy is likely to be involved in the high survival rate recorded in Z10.4-treated plants. This evidences that these two siderophore-producing PGPB induce different strategies in plants to cope with Fe deficiency and to revert chlorosis, while enhancing the fitness of tomato plants.

Gene expression analyses confirmed the different mechanisms triggered by each strain. First, non-inoculated plants show the highest expression in all genes, since plants are activating the mechanisms to absorb iron and reverse chlorosis. However, Z8.8-treated plants in which chlorosis is already reverted show lower expression than controls, suggesting that the PGPB have triggered plant mechanisms to absorb iron, providing a faster and more efficient systemic response involving the transcription of all three genes studied. According to the wave response pattern for systemic responses proposed by Kollist et al. [30], the lower expression of selected genes confirms that this response in transcription has already fallen, and the valley moment has been recorded. On the other hand, it may also be the case that the ability of Z8.8 to produce siderophores that can directly chelate Fe^3+^ results in improved plant nutrition, with no need to stimulate plant *HA1* gene expression, neither in *FRO* nor *IRT* [6,11]. As regards *Pseudomonas* Z10.4, a lower expression was recorded only for the *IRT* gene, showing a reprogramming in plant genes, and speaking of the two mechanisms involved (improved Fe solubility and systemic effect), the dynamics are different, depending on the bacterial species. Altogether, gene expression profiles of control and bacterial-treated plants do not overlap, revealing the gene reprogramming underlying the physiological modifications induced by Z8.8 and Z10.4 [31]. Furthermore, the better results obtained by Z8.8 may be associated with its ability to produce auxins, which will improve the root system, allowing for more efficient absorption due to increased root surface [20].

These results support the hypothesis that the *Pseudomonas* strains Z8.8 and Z10.4 improve plant acquisition of iron, not only via bacterial siderophore-production, but also via a systemic induction of Fe uptake-related genes. Moreover, the wide spectrum of action of these PGPB isolated from a woody species and effective on an herbaceous plant is evidenced. Finally, the ability of these strains to trigger multiple targets to enhance the general fitness of the plant is another fundamental reason to implement the use of PGPB as fertilizers.

All in all, although further studies remain to be conducted to understand the underlying molecular mechanisms, *Pseudomonas* Z8.8 effectively improves plant health more efficiently than Z10.4, and reverts tomato chlorosis by enhancing Fe uptake from soil, confirming its potential to develop organic fertilizers that could be employed in sustainable production.

## 4. Materials and Methods

### 4.1. Isolation of Bacterial Strains

The bacterial strains used in this work were isolated from the rhizosphere of *Pinus pinea* L. grown in a 10 Ha plot, belonging to “El Ejidillo”, a plant nursery in the northern sub-plateau of Spain (coordinates UTM N41°15′33.05″ W3°55′29.96″). According to the USDA classification, the soil has a loamy sand texture (sand: 74.31% ± 1.46; silt: 13.55% ± 1.18; and clay: 12.14 ± 0.51). It has a pH of 7.99 ± 0.03, a nitrogen percentage of 0.125% ± 0.006, an organic matter percentage of 1.58% ± 0.07, and a C/N ratio of 7.72 ± 0.15 [32].

Samples were collected from 10 different zones. Nine plants were sampled in each area. The soil closely adhered to the roots of three plants was pooled and constituted a working unit, all of which were brought to the lab in plastic bags at 4 °C. Rhizosphere soil suspensions were prepared and plated, and individual colonies grown after 24 h were selected, as described by Barriuso et al. [14].

A total of 210 colonies were isolated and phenotypically classified based on morphological traits, Gram-staining and their ability to sporulate. Isolates were maintained in 20% glycerol, at −80 °C.

#### 4.1.1. Metabolic Characterization

Putative beneficial rhizobacteria traits were assayed by different biochemical tests. Phosphate solubilizing ability was measured following the method described by De Freitas et al. [7]; results are expressed as the distance (in mm) from the end of the colony to the end of the hydrolytic halo. Production of auxin-like products was measured following the procedure described by Benizri et al. [33], reporting a positive or negative ability. The ability to produce chitinases was determined as described by Rodríguez–Kábana et al. [34], with modifications [8]. Siderophore production to chelate Fe^3+^ was measured according to Schwyn and Neilands [35]. The sizes of the orange halos in the blue CAS agar were measured after 48 h of bacterial growth, as the distance from the bacteria to the halo border. A selection of strains based on halo sizes ≥ 3 mm was carried out, and the ability to chelate other ions was tested with a modified CAS medium containing manganese (Mn^2+^) and cobalt (Co^2+^) [19].

#### 4.1.2. Genetic Characterization

The 60 strains positive for siderophore production were identified by 16S rRNA gene sequencing. After 24 h of growing in liquid nutrient broth, bacterial DNA was extracted with a genomic DNA isolation kit (Norgen Biotek Corp, Thoroid, ON, Canada). DNA amount and quality were checked with a NanoDrop 2000 (Thermo Scientific, Waltham, MA, USA). Each DNA sample was amplified with 16S rRNA universal primers: 1492R (5′TCGGYTACCTTGTTACGACTT3′) and 27F (5′AGAGTTTGATCMTGGCTCAG3′) [36]. Each primer was added at a final concentration of 0.6 µM with dNTPs (0.25 mM), and the standard reaction buffer with MgCl_2_ and 1.5% of DMSO, and the Biotools Hotsplit polymerase (1 U). Amplification reactions were carried out in a thermocycler (Gene Amp PCR system 2700, Applied Biosystems, South San Francisco, CA, USA) and the temperature was cycled to 94 °C for 2 min and then subjected to 10 cycles, consisting of 94 °C for 0.3 min, 50 °C for 0.30 min, and 72 °C for 1 min, and 20 cycles consisting of 94 °C for 0.3 min, 50 °C for 0.3 min, and 72 °C for 1 min. Finally, the mixtures were incubated at 72 °C for 7 min. Standard sequencing of the PCR products was made by Macrogen (Seoul, Republic of Korea), using the primers described in [36]. The GenBank accession numbers of the 16S rRNA sequences from the 60 isolated strains are shown in Appendix A. Sequence alignment and phylogenetic analyses were carried out on MAFFT v7.0 using the neighbour-joining method, with a bootstrap consensus tree inferred from 1000 replicates.

### 4.2. Biological Assay

From this set of 60 fully characterized PGPB, the two strains Z8.8 (Accession number OP_389294) and Z10.4 (Accession number OP_389298) were selected, to test their ability to revert iron chlorosis in iron-starved *Solanum lycopersicum* L. The selection was carried out considering genetic differences, in vitro ability to produce siderophores, and ability to produce auxins.

### 4.3. Inoculum Preparation

Isolates were maintained in 20% glycerol, frozen at −80 °C, and plated onto Plate Count Agar plates to check viability, incubating them at 28 °C for 24 h. Then, bacterial cells were scraped off the plates into sterile nutrient broth and were incubated for 24 h on an orbital shaker at 28 °C, to obtain a 10^8^ cfu·mL^−1^ inoculum. Culture media were diluted to achieve 10^7^ cfu·mL^−1^, delivering 10 mL per plant in each inoculation.

### 4.4. Experimental Design

Seeds of *Solanum lycopersicum* L. var. Selina were sown in 28-pot trays filled with organic substrate Projar Seed Pro 5050^®^. They were grown under greenhouse conditions from January to March, with natural light photoperiod and temperature between 14 °C and 28 °C. After one month, when plants had developed the third leaf, sodium carbonate buffer (pH 9.2) was applied every three days until the soil pH was stabilized between 8.5 and 8.9, then the pH was measured with a stick-style pH-metre ExStick^®^. At that point, plants showed chlorotic symptoms and were separated into 3 groups, a control and two bacterial treatments (Z8.8 and Z10.4). Each group had 3 replicates, each with 2 trays of 28 plants. A positive control (pH 6 and FeSO_4_) was run in parallel. Then, plants were supplemented with FeCl_3_ in an equivalent iron concentration, as per Hoagland solution recommendations, at the same time as bacterial inoculum. Three bacterial inoculations were delivered by soil drench (10 mL/plant, 10^7^ cfu·mL^−1^) once every 3 days. Nine days after the first inoculation, photosynthesis efficiency was measured as the fluorescence of Photosystem II, as indicated in 4.5. Also, dead, chlorotic, and recovered plants were recorded as the number of dead, yellowish, and greenish plants per treatment. Then, 50% of the plants were dried for nutrient analysis. The remaining plants were separated into three replicates, shoots were harvested for photosynthetic pigments determination, and roots were rinsed and softly dried for gene expression analyses. Plants from each replicate were powdered with liquid nitrogen and kept at −80 °C until further analysis.

### 4.5. Photosynthesis: Chlorophyll Fluorescence Determination

The fluorescence of chlorophyll “a” in Photosystem II was measured with a pulse amplitude modulated (PAM) fluorometer (HansatechFM2, Hansatech, Inc., Norfolk, UK) on 1 h dark-adapted leaves, as described in [10]. The dark-adapted minimum fluorescence (Fo) was measured with weakly modulated irradiation (1 µmol·m^−2^·s^−1^). Maximum fluorescence (Fm) was determined by applying a 700 ms saturating flash (9000 µmol·m^−2^ s^−1^). The variable fluorescence (Fv) was calculated as the difference between the maximum fluorescence (Fm) and the minimum fluorescence (F_0_). The maximum photosynthetic efficiency of Photosystem II (maximal PSII quantum yield) was calculated as Fv/Fm. Immediately, the leaf was continuously irradiated with red–blue actinic beams (80 µmol·m^−2^·s^−1^) and equilibrated for 15 s to record Fs (steady-state fluorescence signal). Following this, another saturation flash (9000 µmol·m^−2^·s^−1^) was applied and then the maximum fluorescence under light-adapted conditions (Fm’) was determined. Other fluorescent parameters were calculated as follows: the effective PSII quantum yield PSII = (Fm’ − Fs)/Fm’ and non-photochemical quenching coefficient NPQ = (Fm − Fm’)/Fm’. Three different plants were measured in the youngest fully expanded leaf.

### 4.6. Photosynthetic Pigments Determination

Photosynthetic pigments content was determined by dissolving one hundred mg of powdered leaves in 1 mL of acetone 80%, vortexed and centrifuged for 5 min at 10,000 r.p.m (Hermle Z233 M-2). Absorbance was measured in 1:10 dilution at 645, 662, and 470 nm in a spectrophotometer (Biomate 5; Thermo Electron Corporation, Waltham, MA, USA) [37].

To calculate chlorophyll “a”, chlorophyll “b”, and carotenoids, the following formulas were used:Chl “a” (µg·g^−1^) = [(12.5 × Abs663) − (2.55 × Abs647)] × V (mL)/weight (g)
Chl “b” (µg·g^−1^) = [(20.31 × Abs647) − (4.91 × Abs663)] × V (mL)/weight (g)
Carotenoids (µg·g^−1^) = ((1000 × Abs470) − (1.82 × Chla) − (85.02 × Chlb))/198) × V (mL)/weight (g)

### 4.7. Plant Nutrient Analysis

Iron (Fe), manganese (Mn), boron (B), and copper (Cu) element content were determined by Mass Spectrometry with Inductively Coupled Plasma (Mass-ICP); cobalt (Co), aluminum (Al), sodium (Na), and zinc (Zn) were analyzed by atomic absorption spectrometry; and chlorine element (Cl) content was determined by volumetry. All those procedures were normalized by Innoagral Laboratories (Seville, Spain), who carried out the analysis. Results are expressed as mg·Kg^−1^ of dry weight (DW).

### 4.8. RNA Extraction and Gene Expression

Before RNA extraction, samples stored at −80 °C were ground to a fine powder with liquid nitrogen, using a sterilized mortar and pestle. Total RNA was extracted from 100 mg of each replicate with a PureLink RNA Mini Kit (Thermo Fisher Scientific; Waltham, MA, USA), adding PVP and DTT as recommended. DNase treatment was also included.

After confirmation of RNA integrity using Nanodrop™, the expression of iron-deficiency responsive genes was analyzed by RT-qPCR. The retrotranscription was performed using an iScript cDNA Synthesis Kit (Bio-Rad, Madrid, Spain) with a GeneAmp PCR System 2700 (Applied Biosystems; Waltham, MA, USA) for 5 min at 25 °C, 30 min at 42 °C, 5 min at 85 °C, and hold at 4 °C. The amplifications were performed with a MiniOpticon Real-Time PCR System (Bio-Rad Laboratories; Madrid, Spain) for 3 min at 96 °C and then 40 cycles consisting of 5 s at 95 °C, 30 s at 60 °C, and a melting curve from 65 °C to 95 °C, with 0.5 °C increments for 5 s. To describe the expression obtained in the analysis, cycle threshold (Ct) was used. Standard curves were calculated for each gene; efficiency values ranged between 80% and 120%. Relative expression (M) was calculated as the 2^−ΔΔCt^ method [38].

The primer sequences for iron nutrition-related genes (H^+^-pump ATPase, *HA1*; reductase, *FRO1*; and membrane Fe II-carrier, *IRT1*) and Actin (housekeeping) are shown in Table 5. Primers were designed with PRIMER3 software, based on the *Solanum lycopersicum* genome.

### 4.9. Statistical Analysis

The homoscedasticity and normality of the variance were previously checked. To determine the statistical differences between experimental treatments, the data from each treatment were compared to the control by a T-Student test. Significant differences were marked with an asterisk (*p* < 0.05). Analyses were performed with Statgraphics Centurion 18 for Windows TM.

## Figures and Tables

**Figure 1 plants-13-03585-f001:**
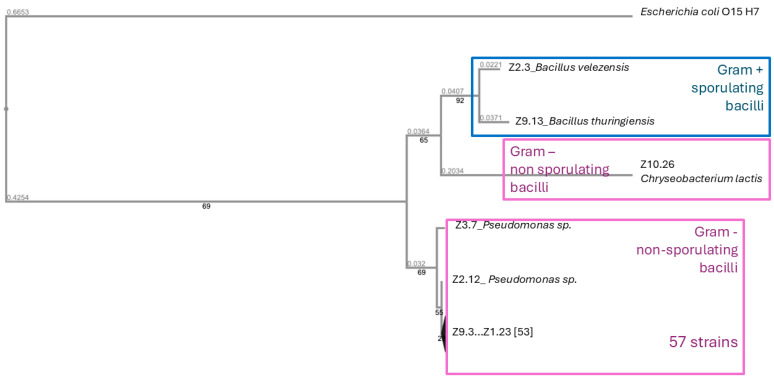
Phylogenetic relationship among siderophore-producing PGPB based on 16S rDNA (60 strains). Neighbour joining was used to infer the evolutionary distances (numbers on the branches) with a bootstrap of 1000 replicates.

**Figure 2 plants-13-03585-f002:**
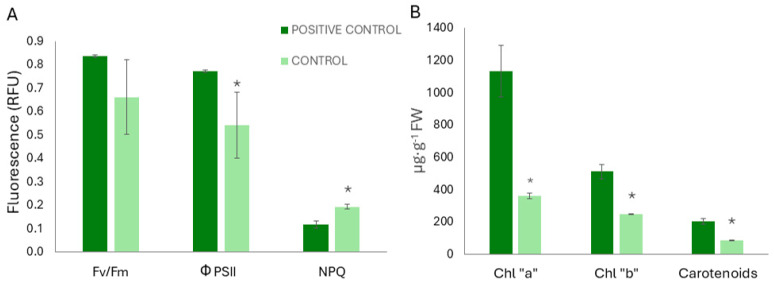
(**A**) Photosynthetic parameters (Fv/Fm; FPSR; NPQ) measured as relative units of fluorescence (RFU). (**B**) Content of photosynthetic pigments (chlorophyll “a”, chlorophyll “b”, and carotenoids), (μg·g^−1^ of FW), in control plants in comparison to positive control plants. Data are expressed as the average (n = 3) ± the standard error. Asterisks show significant differences according to T-Student test (*p* < 0.05).

**Figure 3 plants-13-03585-f003:**
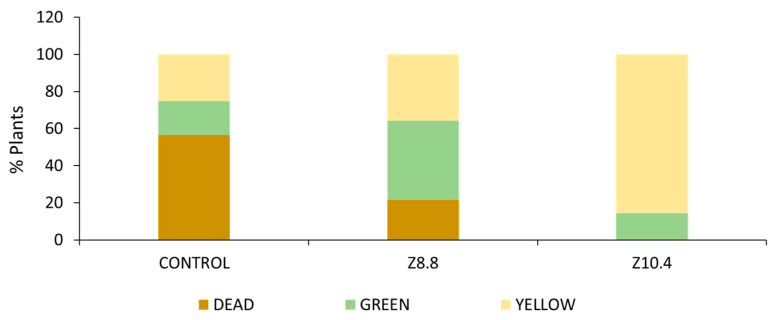
Dead, greenish, and yellow plants (%) in *Pseudomonas* Z8.8- and Z10.4-treated plants and in controls.

**Figure 4 plants-13-03585-f004:**
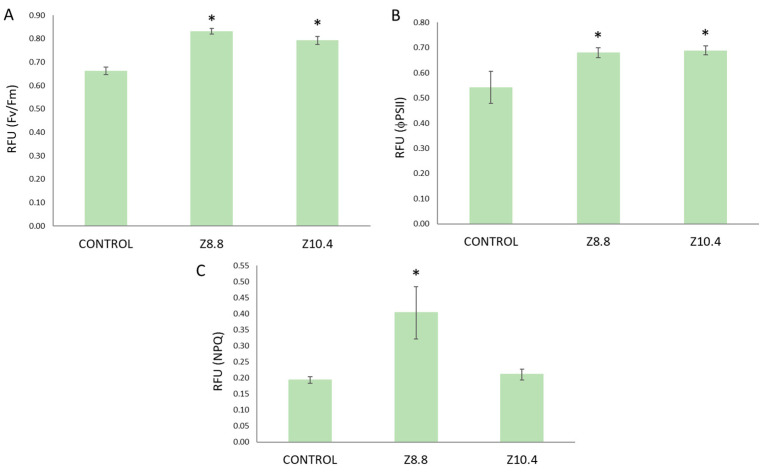
Photosynthetic parameters in inoculated and control plants. (**A**) Fv/Fm; (**B**) FPSII; (**C**) NPQ. RFU (n = 3) are expressed as the average ± the standard error. Asterisks show significant differences with controls according to T-Student test (*p* < 0.05).

**Figure 5 plants-13-03585-f005:**
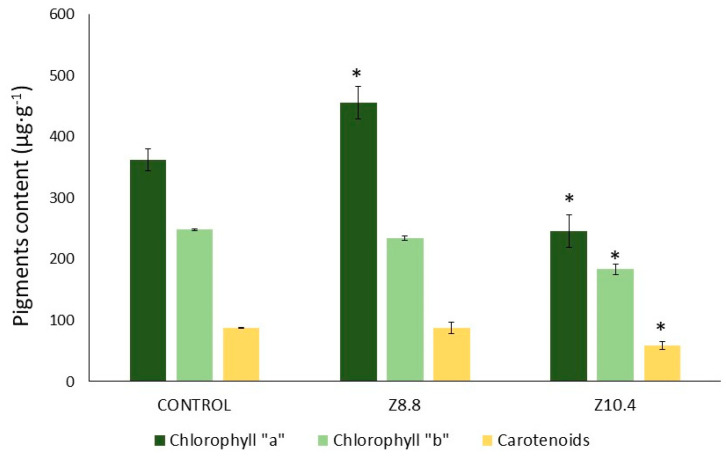
Photosynthetic pigments concentration (μg·g^−1^ fresh weight). Chlorophyll “a”, chlorophyll “b”, and carotenoids measured in control, and Z8.8- and Z10.4-inoculated plants (n = 3). Asterisks show significant differences with controls, according to T-Student test (*p* < 0.05).

**Figure 6 plants-13-03585-f006:**
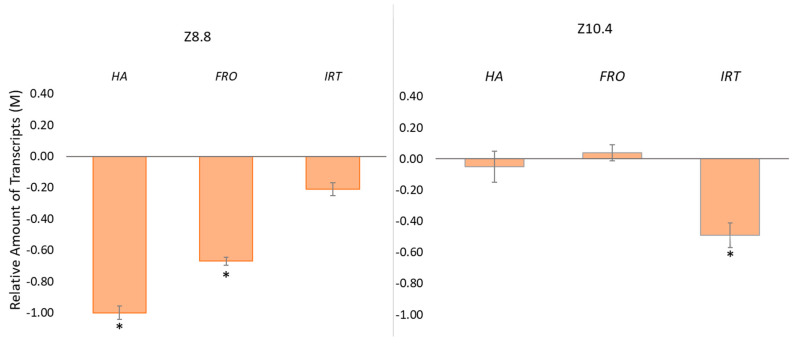
Differential gene expression of iron uptake-related genes (*HA1*; *FRO*; *IRT*) in *Pseudomonas* (Z8.8 and Z10.4)-inoculated roots, expressed as M values with respect to transcript accumulation in control roots (n = 3). Asterisks show significant differences with control according to T-Student test (*p* < 0.05).

**Table 1 plants-13-03585-t001:** Number and percentage of strains with potential PGPB activity.

	Fe Chelators	Mn Chelators	Co Chelators	Phosphate Solubilizers	Auxins Producers	Chitinases Producers
n	60	60	60	19	18	0
%	100	100	100	31.6	30	0

**Table 2 plants-13-03585-t002:** In vitro ion-mobilization potential. n = 3 ± standard error, after 48 h growing in CAS medium.

Strain	CAS-Fe^3+^ mm	CAS-Mn^2+^ mm	CAS-Co^+^ mm
Z8.8	6.1 ± 0.11	7.1± 0.08	5.1 ± 0.08
Z10.4	8.9 ± 0.06	11.0 ± 0.08	8.07 ± 0.09

Potential is expressed as millimetres from the end of the colony to the end of the hydrolytic halo. Fe^3+^: iron; Mn^2+^: manganese; Co^2+^: cobalt.

**Table 3 plants-13-03585-t003:** Nutrient contents in positive control and control plants. Data are expressed as the average (n = 3) ± standard error. Asterisks show significant differences according to T-Student test (*p* < 0.05).

mg·Kg^−1^ DW	Fe	Mn	B	Co	Cu	Zn	Al	Cl	Na
Positive control	5060 ± 559	50.00 ± 3.46	44.00 ± 1.34	0.0243 ± 0.006	67.00 ± 7.50	53.00 ± 6.18	227 ± 22	508 ± 8	382 ± 11
Control	2011 * ± 143	37.07 * ± 0.55	17.87 * ± 0.85	0.018 ± 0.005	53.77 ± 9.61	33.63 ± 2.60	184.00 ± 31.86	517.00 ± 28.54	294.00 ± 67

**Table 4 plants-13-03585-t004:** Nutrient content in Z8.8- and Z10.4-inoculated and in control tomato leaves. Data are expressed as mean (n = 3) ± standard error. Asterisks show significant differences with control (T-Student test: *p* < 0.05).

mg·Kg^−1^ DW	Fe	Mn	B	Co	Cu	Zn	Al	Cl
Control	2011 ± 143	37.07 ± 0.55	17.87 ± 0.85	0.018 ± 0.005	53.77 ± 9.61	33.63 ± 2.60	184.00 ± 31.86	517.00 ± 28.54
Z8.8	2960 * ± 186	38.40 * ± 1.82	18.30 ± 0.42	0.018 ± 0.035	42.80 ± 6.48	27.60 ± 3.33	181.00 ± 5.78	380.00 * ± 16.50
Z10.4	2863 * ± 207	37.40 ± 0.87	12.83 * ± 0.19	0.014 ± 0.023	48.54 ± 5.47	26.93 ± 2.88	183.00 ± 14.73	395.00 * ± 33.23

**Table 5 plants-13-03585-t005:** Primer sequences found in *Solanum lycopersicum* reference sequences.

Gene	Reference Sequence	Left Primer (Fw)	Right Primer (Rv)
*HA1*	NM_001247846.1	5′-cgaaggatagggtcaaacca-3′	5′-agccaccaagaacaactcca-3′
*FRO1*	NM_001247400.2	5′-cctcttcatggttggtcgat-3′	5′-gttcacatggcagaacacg-3′
*IRT1*	NM_001247319.1	5′-gcatcctacaggcggagtataagttc-3′	5′-cagcagcaagaagatcaaccaaagc-3′
*ACTIN*	KY008745.1	5′-ggaaaagcttgcctatgtgg-3′	5′-cctgcagcttccataccaat-3′

## Data Availability

The original contributions presented in this study are included in the article/Appendix A. Further inquiries can be directed to the corresponding author.

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
