# Peer review of "Iron Deficiency in Tomatoes Reversed by Pseudomonas Strains: A Synergistic Role of Siderophores and Plant Gene Activation"

_plants, 2024, doi:10.3390/plants13243585_

Round 1
Reviewer 1 Report (Previous Reviewer 3)
Comments and Suggestions for Authors
Accept in the current files!
Author Response
Thank you very much for your comments. We appreciate your opinion and decision
Sincerely
Reviewer 2 Report (Previous Reviewer 2)
Comments and Suggestions for Authors
Minor observations
The table titles are too long. They must be shorter; any necessary clarifications or details should be noted in the table footnotes.
Table 2. The data for ion mobilization potential are presented with two control variants. These should be explained in the text or in a footnote of the table. The data should be presented as mean + standard deviation. n=3 should be included in the table footnote. The expression (millimeters) of ion mobilization potential should be mentioned in the header table (columns).
Figure 2. Two control variants are not described in the text or in the figure title.
Table 3. The data are expressed in the table (mg·Kg-1) and are also written in the table’s title.

Author Response
Thank you very much for your constructive criticism on the manuscript. Response to minor comments follows below, describing undertaken modifications. However, the comments attached in the pdf file correspond to another ms. We will be pleased to address any more comments if there are any.
"The table titles are too long. They must be shorter; any necessary clarifications or details should be noted in the table footnotes".
We have followed your instructions to shorten table titles removing information to footnotes.
"Table 2. The data for ion mobilization potential are presented with two control variants. These should be explained in the text or in a footnote of the table. The data should be presented as mean + standard deviation. n=3 should be included in the table footnote. The expression (millimeters) of ion mobilization potential should be mentioned in the header table (columns)".
We don’t understand the first 2 sentences about “two control variants” in table 2, unless it refers to table 3, where two controls are presented. In this context, we have explained the two controls for table 3, in the text (L123-124).
Data for table 2 are now presented as mean + SE..
mm have been included in the table, underneath each ion
Table heading has been shortened as: Table 2. In vitro ion mobilization potential (n=3), after 48 hours in CAS medium.
Figure 2. Two control variants are not described in the text or in the figure title.
Control variants are introduced in the text L123-124 and described in “Section 4.4. Experimental design”. A full description was included in the original submission and removed following indications of reviewers and editors. On the other hand, you have called our attention on the length of table and figure legends, so we tried to shorten description.
Nevertheless, positive controls refer to plants with full Hoagland solution and acidic pH on substrate and controls refer to experimental controls with alkaline pH on substrate and FeCl3 as iron source.
Table 3. The data are expressed in the table (mg·Kg-1) and are also written in the table’s title.
We have removed units from the table heading leaving units only the table
This manuscript is a resubmission of an earlier submission. The following is a list of the peer review reports and author responses from that submission.
Round 1
Reviewer 1 Report
Comments and Suggestions for Authors
The article by Montero-Palmero et al., whould be reviewed by the aithors before being accepted for publiaction.
Comments MAIN:
-why are the PGPB selectes from a Pinus pinea? What is the link between this plant and iron dificiency?
-Are there any specific characteristic of the soil where Pinus pinea grows? If yes, please specify the soil type!
- a "control" is usually aplant that grows under perfect balance conditions, thinking of the "control" as the iron-deficiency condition is very confusing
-in the materials and methods it is written that 5 type of conditions were used (4.4), why are the results anly reported for 3 of them? if you don't compare with the right controls, how can you be sure that what you see is linked tothe PGPB and not to something else?
-summplementary materials are missing
-what's the itron concentration in the different treatments? This is a key aspect that was not measured
- table 1: are the chracateristics of Fe, Mn and Co shared by the same 60 microbes?
-figures should have a better contrast, as some of them are difficult to read, also text in figures is not alwasy easy to read
MINOR:
when using acrocynms, these should be defined ONLY the first time, from then on, only acronyms should be used. The text is a mix of both
Comments on the Quality of English LanguageThe text contains several typos and for some sentences english sould be reviewed (check for example the "p" of P-value that should not be capital, chemical formulas, organisms name not in italic...
Author Response
Comments MAIN:
-why are the PGPB selectes from a Pinus pinea? What is the link between this plant and iron dificiency? -
We agree with the reviewer that the connection between the PGPB isolated and the source was not evident. For that purpose, we have included the information below in the revised version, L67-L70
Pinus pinea is a mediterranean plant, that usually grows in sandy nutrient-poor soils. Hence, we hypothetized that under these conditions, Pinus would select specialized strains to cooperate in nutrient acquisition. This hypothesis has been already contrasted by our group on a previous PGPR screening in Pinus (Barriuso et al, 2005, doi.10.1007/s00248-004-0112-9) that resulted in several strains being effective for plant growth (Barriuso et al, 2008 doi:10.1094 / PHYTO-98-6-0666,
Barriuso et al, 2008. doi:10.1111/j.1365-2672.2008.03862.x) and enhanced defense (Barriuso et al 2008 doi: 10.1111/j.1467-7652.2008.00331.x).
Are there any specific characteristic of the soil where Pinus pinea grows? If yes, please specify the soil type!
According to the USDA classification, the soil has a loamy sand texture (sand: 74.31% ± 1.46; silt: 13.55% ± 1.18; clay: 12.14 ± 0.51). It has a pH of 7.99 ± 0.03, a nitrogen percentage of 0.125% ± 0.006, an organic matter percentage of 1.58% ± 0.07, and a C/N ratio of 7.72 ± 0.15.
This information has been included in the Materials and Methods section, L300-L303 of the revised version.
- a "control" is usually aplant that grows under perfect balance conditions, thinking of the "control" as the iron-deficiency condition is very confusing. in the materials and methods it is written that 5 type of conditions were used (4.4), why are the results anly reported for 3 of them? if you don't compare with the right controls, how can you be sure that what you see is linked tothe PGPB and not to something else?
We apologize for this lack of clarity. We have rewritten the experimental set up in order to improve understanding; the 5 experimental conditions have been deleted and we refer to a positive control with full nutrient delivery and the 3 treatments under evaluation: control and 2 strains.
It now reads:
“Seeds of Solanum lycopersicum L. var. Selina were sown in a 28-pot tray filled with organic substrate Projar Seed Pro 5050®. They were grown under greenhouse conditions from January through March, with natural light photoperiod and temperature between 14°C and 28°C. After one month, when plants had developed the third leaf, sodium carbonate buffer (pH 9.2) was applied every three days until soil pH was stabilized between 8.5-8.9; pH was measured with a stick-style pH-metre ExStick®. At that point, plants showed chlorotic symptoms, and separated in 3 groups, a control and 2 bacterial treatments (Z8.8 and Z10.4); each group had 3 replicates, each with 2 trays of 28 plants. Then, plants were supplemented with FeCl3 in an equivalent iron concentration as Hoagland solution recommendations, at the same time as bacterial inoculum. Three bacterial inoculations were delivered by soil drench (10 mL/plant, 107 cfu·mL−1), once every 3 days. Nine days after the first inoculation, photosynthesis efficiency was measured as the fluorescence of Photosystem II. Also, dead, chlorotic, and recovered plants were recorded as the number of dead, yellowish and greenish plants per treatment. Then, 50% of the plants were dried for nutrient analysis. The remaining plants were separated into three replicates; shoots were harvested for photosynthetic pigments determination; roots were rinsed and softly dried, for gene expression analyses. Plants from each replicate were powdered with liquid nitrogen and kept at -80°C until further analysis. A positive control in which plants grew under non-stressed conditions (acidic pH and FeSO4 (n=28) was grown in parallel to prove effects of Fe deficiency”
-summplementary materials are missing
We apologize for this inconvenience. The supplementary file was not uploaded probably due to an informatic missfunction. It has been uploaded now.
-what's the itron concentration in the different treatments? This is a key aspect that was not measured
We are not sure of the question at this point.
If you wonder how much iron was added to plants, we have stated in L358-359 of the original version that the amount of iron provided is equivalent to the concentration of Hoagland solution.
If you refer to the concentration of iron found in plants with the different bacterial treatments, this information is included in Table 4. Table 3 shows a comparation between the positive controls and experimental controls, and show a lower iron concentration, therefore validating experimental conditions to proof the ability of strains to provide iron to plants.
- table 1: are the chracateristics of Fe, Mn and Co shared by the same 60 microbes?
Yes, all 60 strains are able to mobilize Fe, Mn and Co, that is why they were selected from the pool of 210 strains under study. This information is shown in Table 2 supplementary material
-figures should have a better contrast, as some of them are difficult to read, also text in figures is not always easy to read
Figure quality has been improved in order to make them more easy to read.
MINOR:
when using acrocynms, these should be defined ONLY the first time, from then on, only acronyms should be used. The text is a mix of both
Acronyms have been reviewed as requested, and modifications done throughout the text.
Comments on the Quality of English Language
The text contains several typos and for some sentences english sould be reviewed (check for example the "p" of P-value that should not be capital, chemical formulas, organisms name not in italic...
Text has been carefully reviewed by a native English speaking colleague.
Reviewer 2 Report
Comments and Suggestions for Authors
Review_v1 for manuscript: plants-3272917
Reverting Chlorosis in iron-starved tomato by two PGPB Pseudomonas strains involves a combination of bacterial siderophores and a systemic induction of Fe-uptake responsive genes.
Authors
MARIA BELEN MONTERO PALMERO , Jose Antonio Lucas , Blanca MONTALBÁN , Ana GARCÍA-VILLARACO , Javier Gutierrez-Mañero , Beatriz Ramos-Solano
The purpose of this study is clearly defined as using PGPB to enhance Fe nutrition and improve plant capability in soils with low soluble Fe. PGPBs are widely known as biostimulants or biofertilizers for plant nutrient assimilation and ensuring crop quality.
The first goal was identifying bacterial strains that could improve tomato plant nutrition, focusing on iron nutrition in alkaline soils. Two actions were taken to this end: selecting siderophore-producing bacteria from a large set of isolated PGPB by sequencing 16S rDNA and evaluating the ability of the best siderophore-producing strains (Z8.8 and Z10.4) to revert chlorosis in Fe-starved tomato plants.
I found the paper to be well written overall. The analysis methods and the experimental design are exposed correctly and in detail.
The results are processed statistically and presented graphically, with averages and the significance of the differences. The discussions are detailed, clearly expressed, and scientific, following similar research.
The bibliographic resources are under the research carried out, relevant and are of recent date.
I cannot recommend this manuscript for publication in this form because Table 1's supplementary data is missing.
In addition, I have minor observations:
Line 329 - The inoculum density must be corrected.
What culture medium did you use for auxin production potential for bacterial isolate traits? It is not specified in the paper.
You used several controls in the experiment. I suggest using their different abbreviations to help understand the data more easily.
Author Response
Reverting Chlorosis in iron-starved tomato by two PGPB Pseudomonas strains involves a combination of bacterial siderophores and a systemic induction of Fe-uptake responsive genes.
Authors
MARIA BELEN MONTERO PALMERO , Jose Antonio Lucas , Blanca MONTALBÁN , Ana GARCÍA-VILLARACO , Javier Gutierrez-Mañero , Beatriz Ramos-Solano
The purpose of this study is clearly defined as using PGPB to enhance Fe nutrition and improve plant capability in soils with low soluble Fe. PGPBs are widely known as biostimulants or biofertilizers for plant nutrient assimilation and ensuring crop quality.
The first goal was identifying bacterial strains that could improve tomato plant nutrition, focusing on iron nutrition in alkaline soils. Two actions were taken to this end: selecting siderophore-producing bacteria from a large set of isolated PGPB by sequencing 16S rDNA and evaluating the ability of the best siderophore-producing strains (Z8.8 and Z10.4) to revert chlorosis in Fe-starved tomato plants.
I found the paper to be well written overall. The analysis methods and the experimental design are exposed correctly and in detail.
The results are processed statistically and presented graphically, with averages and the significance of the differences. The discussions are detailed, clearly expressed, and scientific, following similar research.
The bibliographic resources are under the research carried out, relevant and are of recent date.
I cannot recommend this manuscript for publication in this form because Table 1's supplementary data is missing.
We apologize for this inconvenience. The supplementary file was not uploaded probably due to an informatic missfunction. It has been uploaded now.
In addition, I have minor observations:
Line 329 - The inoculum density must be corrected.
This typo has been corrected. Thanks for the observation. It’s now in line 347-348 of the revised version
What culture medium did you use for auxin production potential for bacterial isolate traits? It is not specified in the paper.
Auxin potential trait was based on the methodology proposed by Benizri et al. [11], as indicated in L319 of original version. Yeast glucose medium (5 g yeast extract, 10g glucose) was used to grow bacteria; after removing bacterial by centrifugation and 0.25 mm filtration, 3 mL were mixed with 6 mL Slakowsky reagent (FeCl3 0.5M 2% in perchloric acid 35%). Absorbance was measured at 530nm. Data was interpolated on a calibration curve of IAA; it was considered positive when more than 2 ppm were produced.
You used several controls in the experiment. I suggest using their different abbreviations to help understand the data more easily.
We apologize for this lack of clarity, which has also been raised by reviewer 1. We have rewritten the experimental set up in order to improve understanding; the 5 experimental conditions have been deleted and we refer to a positive control with full nutrient delivery and the 3 treatments under evaluation: control and 2 strains.
It now reads:
“Seeds of Solanum lycopersicum L. var. Selina were sown in a 28-pot tray filled with organic substrate Projar Seed Pro 5050®. They were grown under greenhouse conditions from January through March, with natural light photoperiod and temperature between 14°C and 28°C. After one month, when plants had developed the third leaf, sodium carbonate buffer (pH 9.2) was applied every three days until soil pH was stabilized between 8.5-8.9; pH was measured with a stick-style pH-metre ExStick®. At that point, plants showed chlorotic symptoms, and separated in 3 groups, a control and 2 bacterial treatments (Z8.8 and Z10.4); each group had 3 replicates, each with 2 trays of 28 plants. Then, plants were supplemented with FeCl3 in an equivalent iron concentration as Hoagland solution recommendations, at the same time as bacterial inoculum. Three bacterial inoculations were delivered by soil drench (10 mL/plant, 107 cfu·mL−1), once every 3 days. Nine days after the first inoculation, photosynthesis efficiency was measured as the fluorescence of Photosystem II. Also, dead, chlorotic, and recovered plants were recorded as the number of dead, yellowish and greenish plants per treatment. Then, 50% of the plants were dried for nutrient analysis. The remaining plants were separated into three replicates; shoots were harvested for photosynthetic pigments determination; roots were rinsed and softly dried, for gene expression analyses. Plants from each replicate were powdered with liquid nitrogen and kept at -80°C until further analysis. A positive control in which plants grew under non-stressed conditions (acidic pH and FeSO4 (n=28) was grown in parallel to prove effects of Fe deficiency”
Reviewer 3 Report
Comments and Suggestions for Authors
The manuscript of “Reverting Chlorosis in iron-starved tomato by two PGPB Pseudomonas strains involves a combination of bacterial siderophores and a systemic induction of Fe-uptake responsive genes” did many works. Authors selected siderophore-producing bacteria isolated from the rhizosphere of Pinus pinea L, evaluated the ability of the best two strains to revert chlorosis in Fe-starved tomato plants, found that Pseudomonas Z8.8 had many nice characters, it could be effective candidate to develop biofertilizers. The manuscript is interesting, but there were many questions existed in the present study.
1. The title is too long to boring, I suggest that authors simplify the title to make it more attractive.
2. The abstract cost many sentences to introduce the background of this study, and didn’t perfectly summarize the results and highlights of this manuscript, please supply to make the abstract more perfect. Especially, in results, this manuscript analyzed many content, but the abstract didn’t summarized these important and essential results found in this manuscript, so I suggest that author should updated the abstract.
3. there were too much grammar mistakes involved in this study, please polish the language of this manuscript.
4. I suggest that author add Pseudomonas to the keywords.
5. the introduction is poorly without charming, and only eight references involved in the introduction, I think the number of references is too little to explain successfully the underground of this study, please increase the quality of introduction.
6. the figure 1, I suggest that author add explanation for the line, because many lines were long, and other were short.
7. the title of table 4 should be removed to the above of this table.
8. the discussion should be revised, many sentences for summarizing results of this manuscript in discussion were similar or same with the results, which let the discussion without attractive.
9. please check these references, including these references is or isn’t involved in this study, and authors’ name, journals, and more detailed information.
Author Response
The manuscript of “Reverting Chlorosis in iron-starved tomato by two PGPB Pseudomonas strains involves a combination of bacterial siderophores and a systemic induction of Fe-uptake responsive genes” did many works. Authors selected siderophore-producing bacteria isolated from the rhizosphere of Pinus pinea L, evaluated the ability of the best two strains to revert chlorosis in Fe-starved tomato plants, found that Pseudomonas Z8.8 had many nice characters, it could be effective candidate to develop biofertilizers. The manuscript is interesting, but there were many questions existed in the present study.
- The title is too long to boring, I suggest that authors simplify the title to make it more attractive.
We regret to hear that it was boring. Let’s hope that our new proposal is fun and more attractive. Otherwise, we would love a suggestion. The new title is: Reverting iron deficiency in tomatoes by Pseudomonas strains involve bacterial siderophores and plant gene activation.
- The abstract cost many sentences to introduce the background of this study, and didn’t perfectly summarize the results and highlights of this manuscript, please supply to make the abstract more perfect. Especially, in results, this manuscript analyzed many content, but the abstract didn’t summarized these important and essential results found in this manuscript, so I suggest that author should updated the abstract.
We have rewritten the abstract to better gather results obtained.
- there were too much grammar mistakes involved in this study, please polish the language of this manuscript.
We regret to hear this. We have undertaken a thorough revision.
- I suggest that author add Pseudomonasto the keywords.
Thank you very much for the suggestion. We have included “Pseudomonas” to keywords. In addition to including “Pseudomonas” to keywords, “PGPB” has been spelled out and “photosynthesis” has also been included.
- the introduction is poorly without charming, and only eight references involved in the introduction, I think the number of references is too little to explain successfully the underground of this study, please increase the quality of introduction.
We have included a new item to our Introduction about Pinus. Also, we have reviewed and included more references to widen scope of the Introduction.
- the figure 1, I suggest that author add explanation for the line, because many lines were long, and other were short.
The following has been added to the leyend of Figure 1: Phylogenetic relationship among siderophore-producing PGPB based on 16S rDNA (60 strains). Neighbour joining was used to infer the evolutionary distances (numbers on the branches) with a bootstrap of 1000 replicates.
- the title of table 4 should be removed to the above of this table.
If we understand correctly, the title “nutrient content” should only be on the table legend. We have undertaken this movement. If this is not properly understood, we will be glad to make further modifications. We have done the same on Table 3.
- the discussion should be revised, many sentences for summarizing results of this manuscript in discussion were similar or same with the results, which let the discussion without attractive.
We have revised the discussion in order to make it easier to read and more friendly.
- please check these references, including these references is or isn’t involved in this study, and authors’ name, journals, and more detailed information.
We have carefully reviewed the reference list to match the references listed in the article.
Round 2
Reviewer 3 Report
Comments and Suggestions for Authors
Thanks for the revision, and all the comments were revised!
Author Response
Thank you.